# Physical Properties and the Reconstruction of Unstable Decahedral Silver Nanoparticles Synthesized Using Plasmon-Mediated Photochemical Process

**DOI:** 10.3390/nano12071062

**Published:** 2022-03-24

**Authors:** Jui-Chang Chen, Yu-Te Chu, Shi-Hise Chang, Ya-Tin Chuang, Cheng-Liang Huang

**Affiliations:** Department of Applied Chemistry, National Chiayi University, Chiayi City 600355, Taiwan; chenjc@mail.ncyu.edu.tw (J.-C.C.); mic0000@hotmail.com (Y.-T.C.); s0982723@mail.ncyu.edu.tw (S.-H.C.); apple220191@gmail.com (Y.-T.C.)

**Keywords:** plasmon-mediated, decahedral silver nanoparticles, silver nanoprisms, SERS, shape transformation, reconstruction

## Abstract

Plasmon-mediated shape transformation from quasi-spherical silver nanoparticles (AgNPs) to silver nanoprisms (AgNPrs) and decahedral silver nanoparticles (D-AgNPs) under irradiation of blue LEDs (λ = 456 ± 12 nm, 80 mW/cm^2^) was studied at temperatures ranging between 60, 40, 30, 20, 10, and 0 °C. It was found that reaction temperature affected transformation rates and influenced the morphology distribution of final products. The major products synthesized at temperatures between 60 °C and 0 °C were AgNPrs and D-AgNPs, respectively. The D-AgNPs synthesized at such low temperatures are unstable and become blunt when light irradiation is removed after the photochemical synthesis. These blunt nanoparticles with pentagonal multiple-twinned structures can be further used as the seeds to reconstruct complete D-AgNPs after irradiating blue LEDs at various bath temperatures. Our results showed that these rebuilt D-AgNPs are much more stable when at higher bath temperatures. Furthermore, the rebuilt D-AgNPs (edge lengths ~41 nm) can grow into larger D-AgNPs (edge lengths ~53 nm) after the irradiation of green LEDs. Surface-enhanced Raman spectra of CV in AgNP colloids showed that D-AgNP colloids have better SERS enhancements factors than AgNPrs.

## 1. Introduction

Silver nanoparticles (AgNPs) have been applied in the fields of biosensing [1,2], optoelectronic devices [3,4], and the active substrates for SERS [5,6,7,8,9,10,11,12,13,14,15,16,17,18,19], due to their wealth of optical properties mainly originating from light absorption and scattering by the localized surface plasmon resonance (LSPR) [20,21]. The frequency of LSPR, which is associated with the collective oscillation of particles’ free electrons in the conduction band, is strongly dependent on the size, shape, composition, crystallinity, interparticle spacing, and local dielectric environment [22]. Among these factors, shape-controlled synthesis has been proven to be one of the most effective methods for fine-tuning the optical properties and functions of metal nanostructures [23,24,25]. AgNPs with different shapes, such as rods [26,27,28], cubes [29], shells [30], tetrahedrons [31], triangular bipyramids [32], decahedra [33,34,35,36,37,38,39,40,41], and prisms [42,43,44,45] have been synthesized.

Shape-controlled synthesis of NPs in solution via photochemical reactions can be traced easily by measuring the time-resolved UV–vis spectra and TEM images since the reaction can be stopped rapidly by removing the light irradiation [34,35,46]. Various surfactants and polymers are often used in photochemical reactions to obtain NPs of desired shapes because of their preferential adsorption to specific facets, leading to anisotropic growth rates [28]. In addition, different wavelengths of the irradiation light can also be chosen to synthesize the nanostructures, resulting in varying morphologies [46,47]. Mirkin et al. have reported that the shapes of products in a photochemical growth process are more anisotropic than those in the thermal growth process [40]. However, not much is known about the effects of temperature on the photo-induced shape transformation of AgNPs, especially at temperatures around the freezing point of water [33,34,36].

In this study, we found that the major products of the spherical AgNP colloids irradiated with blue LEDs (456 ± 12 nm) at 60 °C were AgNPrs, consistent with our previous studies [34,36]. However, the major products became decahedral silver NPs at temperatures lower than 10 °C. The tips and edges of decahedral silver NPs synthesized by this method at such a low temperature undergo a decay process to become blunt after removing the light for 30 min. Compared to previous studies [33,35], the decahedral silver NPs synthesized using this present method are less stable than those synthesized using other methods. This unusual fragile property possibly results from the shape conversion performed under a relatively low incubation temperature [36]. After the decay, the round AgNPs still have multiple twinning structures [36]. The round AgNPs with multiple twinning structures can be used as the seeds to reconstruct D-AgNPs with sharp corners under the blue LED irradiation. D-AgNPs rebuilt with a higher bath temperature (i.e., 60 °C) show higher stability than those rebuilt with a lower bath temperature (i.e., 0 °C). Furthermore, these D-AgNPs can increase their sizes via an enlargement process under green LED irradiation. The measurement of CV SERS spectra shows that D-AgNP colloids have much higher SERS enhancement factors than the AgNPr colloids.

## 2. Materials and Methods

### 2.1. Materials

Silver nitrate was purchased from Showa Chemical Co. (Tokyo, Japan). Sodium citrate, sodium tartrate, and potassium bromide were purchased from J. T. Baker (J.T. Baker, Philipsburg, NJ, USA). Sodium borohydride, hydrogen peroxide, PVP (Mw = 40,000), and 16-mercaptohexadecanoic (16-MHDA) were purchased from Sigma-Aldrich (Sigma-Aldrich, Inc., St. Louis, MO, USA). All chemicals were of analytical grade and were used without further purification. Milli-Q grade water (>18 MΩ) was used in all experiments.

### 2.2. Instrumentation

A Joel JEM-2100 (Japan Electron Optics Laboratory Co., Ltd., Tokyo, Japan) transmission electron microscope (TEM) was employed to obtain TEM images of each sample. The TEM instrument was operated at 100 KV or 200 KV. Before analysis by TEM, 1 mL aliquots of silver colloids were added to 10 μL of 10^−3^ M 16-MHDA to prevent further shape transformation, then dripped onto a carbon-coated copper grid, and air-dried at room temperature. All UV–vis extinction spectra were recorded at 25 °C on a Hitachi U-2800 spectrophotometer (Hitachi Science & Technology, Tokyo, Japan) using a quartz cuvette with an optical path of 10 mm. Powder X-ray diffraction (XRD) was performed on a Bruker D8 Discover X-ray diffractometer with a Cu K radiation source, and the sample was deposited onto a glass slide.

### 2.3. Colloids Preparation

#### 2.3.1. Synthesis of AgNPs Using the Plasmon-Mediated Method at Different Temperatures

AgNPs were prepared using a plasmon-mediated photochemistry process [34,36]. In a typical synthesis, AgNP seeds were prepared by adding 1 mL of 1.0 × 10^−2^ M silver nitrate and 1 mL of 3.0 × 10^−2^ M sodium citrate into 97 mL DI water under vigorous stirring and then by adding 1 mL of 3.0 × 10^−2^ M sodium borohydride dropwise. A yellow-colored solution was observed, corresponding to quasi-spherical NPs with diameters of 5–10 nm that were formed. The solution was subsequently irradiated with blue LEDs (λ = 456 ± 12 nm, 80 mW/cm^2^). For temperature-controlled experiments, the vial with the mixture was placed in a chamber connected with a temperature-controlled water circulating system and was irradiated under the blue LEDs. Appendix A shows the setup of the temperature-controlled light irradiation system with blue LEDs [35,36].

#### 2.3.2. Reconstruction and Enlargement of D-AgNP Colloids Using the Plasmon-Mediated Method at Different Temperatures

Decahedral silver nanoparticles (D-AgNPs) synthesized using the plasmon-mediated process at lower temperatures (0 and 10 °C) were not stable and could not maintain their original morphology for a long period. The corners and edges of D-AgNPs would decay and become blunt within 30 min. However, these blunt D-AgNPs can be used as seeds to rebuild much more stable and complete D-AgNPs with sharp corners under blue LED irradiation (80 mW/cm^2^) for the reconstruction process. Bath temperatures of the reconstruction process were controlled at 0, 10, 20, 30, 40, 50, 60, and 70 °C. The irradiation times of the reconstruction process at 0, 10, 20, 30, 40, 50, 60, and 70 °C are 6, 5, 4, 3.5, 2, 1, and 0.8 h, respectively. After the reconstruction process (with blue LEDs), the AgNP colloids were further irradiated with green LEDs (520 ± 20 nm, 35 mW/cm^2^; this is called the enlargement process) for 3 h (for the enlargement process). In short, stable small D-AgNPs can be synthesized using the plasmon-mediated process in three stages: (1) formation of D-AgNPs at a lower bath temperature; (2) decay process when light irradiation is removed; and (3) reconstruction process at a higher bath temperature. Stable large D-AgNPs can be formed when the stable small D-AgNPs are irradiated under green LEDs for 3 h.

### 2.4. SERS Measurement

The SERS spectra of crystal violet (CV) were recorded using the AgNP colloids (AgNPr colloids, small D-AgNP colloids, and large D-AgNP colloids) synthesized in this study. A solution of CV (10^−5^ M, 0.1 mL) was added to 0.9 mL of each AgNP colloid for a typical measurement [35]. Prepared samples were placed in a quartz cuvette and excited by a 20 mW laser beam at 532 nm. The regular acquisition time of a CV SERS spectrum was 30 s.

## 3. Results

### 3.1. Synthesis of Silver NP Colloids at Different Temperatures

Figure 1a–f show the time-dependent UV–vis spectra of silver NP colloids synthesized using the plasmon-mediated method under blue LEDs at 60, 40, 30, 20, 10, and 0 °C, respectively. The irradiation times to reach the maximum intensities corresponding to LSPR peaks of nanostructures at 60, 40, 30, 20, 10, and 0 °C were 60, 100, 160, 270, 360, and 420 min, respectively. This observation indicates that elevated reaction temperatures increase the rate of shape transformation.

### 3.2. Characterization of Temperature-Dependent Silver NP Colloids by LSPR Spectra and TEM Images

Figure 2 shows the spectra A-F corresponding to the product nanostructures synthesized at 60, 40, 30, 20, 10, and 0 °C, respectively. According to the previous results [22,34,43], peaks at 337, 410, and 500 nm in curve A (60 °C) of Figure 2 correspond to out-of-plane quadrupole, in-plane quadrupole, and in-plane dipole LSPR modes of AgNPrs, respectively. The results revealed that the major products of nanostructures are AgNPrs at higher temperatures, indicated by the distinct peak at 337 nm.

Figure 3a shows the TEM image of the silver nanostructures synthesized at 60 °C. Most of the nanostructures were triangular, with a broad size distribution ranging from ca. 20 nm to 90 nm in edge length. The most intense LSPR bands in curves A–F of Figure 2 all peaked in the range of 486–495 nm. However, the relative intensities of out-of-plane quadrupole LSPR mode of AgNPrs peaked at about 333 nm and decreased at lower temperatures (see the inset of Figure 2). This result indicates that the ratio of AgNPrs to the nanostructures with different shapes becomes smaller at lower temperatures. Figure 3b–f show the TEM images corresponding to the products synthesized at 40, 30, 20, 10, and 0 °C, respectively. The AgNPrs and the D-AgNPs can be observed as well. Furthermore, the ratio (D-AgNPs/AgNPrs) increased as temperatures decreased. The major products at bath temperatures less than 20 °C are D-AgNPs with edge lengths between 30 and 55 nm.

Figure 4 shows the percentages of decahedral silver NPs in the final products synthesized at varying temperatures. All of the data points with the standard deviations in the diagram were estimated from the TEM images of the nanostructures synthesized at least three times. It can be concluded that photo-induced shape conversion reaction favors AgNPrs at higher temperatures while favoring decahedral silver NPs at lower temperatures. This result is consistent with several groups’ previous studies, as well as our own [34,36,38].

### 3.3. The Instability of D-AgNPs Synthesized at a Lower Temperature

Figure 5 shows the digital photographs and the corresponding UV–vis spectra of D-AgNPs at an ambient temperature after the photochemical reaction. Within 30 min, the color of the silver colloid changed from orange to yellow, corresponding with the observation that the LSPR band blue shifted from 487 nm toward 448 nm.

This blue shift corresponds to the snipping of corners and edges of D-AgNPs [36]. Figure 6 shows the TEM image of these snipped D-AgNPs. This observed instability was not detected in the D-AgNPs synthesized using other methods in the previous studies because these as-prepared D-AgNPs lacked an annealing process when synthesized at low temperatures. Heat treatment can improve the elasticity of bulk metal materials because atoms can rearrange to reduce dislocations in the crystals. Thus, the instability of these D-AgNPs might result from the defects inside these nanostructures because silver atoms do not have enough kinetic energy to migrate at a relatively low bath temperature. To prevent the shape evolution before the TEM measurements, 10^−5^ M 16-MHDA was added to the silver colloids, except in Figure 6 in this study.

### 3.4. Reconstruction of D-AgNPs at Different Temperatures

Figure 7a,b show the time-dependent extinction spectra of silver colloids under the irradiation of blue LED for the reconstruction process using the blunt D-AgNPs as the seeds (as shown in Figure 6) at 10 °C and 60 °C, respectively. Though the rate in the reconstruction process is at 10 °C and is slower than that at 60 °C, the time evolutions of spectra at these two temperatures are similar. Both spectra at the completed stage (6 h for 10 °C and 60 min for 60 °C) show a strong peak at 490–495 nm (the longitudinal dipolar LSPR band for D-AgNPs) and a weak peak at 395–410 nm (the transverse dipolar LSPR band for D-AgNPs), without a peak at 330–340 (corresponding to the out-of-plane quadrupole LSPR band of AgNPrs). These spectroscopic observations indicate that the products at different temperatures are D-AgNPs.

Figure 8a,b show the TEM images of AgNPs after the reconstruction processes at 10 °C and 60 °C for 6 h and 60 min, respectively. The TEM observation that most of the AgNPs are decahedra with edge lengths of approximately 30–55 nm is consistent with the spectra in Figure 7. However, the XRD spectra show that the D-AgNPs rebuilt at 10 °C and 60 °C have different grain sizes, according to the different bandwidths (shown in Appendix A).

The stability of D-AgNPs reconstructed at different bath temperatures is different. Figure 9 shows the time-dependent wavelength of the longitudinal dipolar LSPR band of D-AgNPs (at approximately 490 nm) after the reconstruction processes at different bath temperatures. The LSPR wavelength of D-AgNPs reconstructed at 50, 60, and 70 °C almost remained unchanged, while the LSPR wavelength of D-AgNPs gradually rebuilt at 0–40 °C blue shift after removing the light source. D-AgNPs reconstructed at higher temperatures (50, 60, and 70 °C) are much more stable than D-AgNPs rebuilt at lower temperatures (0–40 °C). TEM images (Figure 9 inset) confirm this observation. These results indicate that the higher average thermal energy (at higher bath temperature) benefits from annealing to reduce the defects inside the nanostructures and to promote their stability in this reconstruction process.

### 3.5. Increase the Size of D-AgNPs

Figure 10 shows the TEM image of AgNPs reconstructed at 60 °C under blue LEDs for 1 h (using the blunt D-AgNPs as the seeds, as shown in Figure 6), followed by the irradiation under green LEDs ($\lambda_{\max}=520\pm 20\;\mathrm{nm}$, 35 mW/cm^2^) for 3 h. Most AgNPs are decahedra with 45–60 nm edge lengths. Thus, after the green LED irradiation process, D-AgNPs increase the edge lengths by more than 30%.

Two different sizes of stable D-AgNPs are synthesized. Appendix A shows the histogram of particle sizes after the reconstruction and enlargement processes. The average edge lengths of D-AgNPs are 40.8 ± 5.1 nm and 53.3 ± 4.0 nm for the reconstruction and enlargement processes, respectively. Figure 1 shows the formation, decay, reconstruction, and enlargement of D-AgNPs synthesized using a plasmon-mediated process. In short, D-AgNPs can be synthesized using a plasmon-mediated process at 0 °C under blue LEDs for 7 h. The corners of the D-AgNPs will decay if light irradiation is stopped. The instability results from the defects inside the nanocrystal structure. After the decay, the AgNPs with multiple twinning structures can be used as seeds to rebuild D-AgNPs under blue LEDs at 60 °C for 1 h. The small D-AgNPs can grow to larger D-AgNPs under green LEDs at 60 °C for 3 h.

### 3.6. Optical Property of Large D-AgNPs

The D-AgNP colloids prepared by these processes show the bicolor properties [33]. Figure 11 shows the pictures of D-AgNP colloids in a vial taken from different view angles. The colloidal D-AgNPs show red (or pink) color when the background is white but green when the background is black. The red color is related to extinction, and the green is related to scattering. The similar behaviors of Lycurgus cup and D-AgNP colloids indicate the strongly scattering behavior of larger metal nanoparticles [40,48].

### 3.7. Application for SERS

Figure 12 shows the SERS spectra of 1 × 10^−6^ M crystal violet (CV) excited with a 532-nm laser with different AgNP colloids (AgNPr colloids, small D-AgNP colloids, and large decahedral silver NP colloids). These SERS spectra show that decahedral AgNP colloids using the reconstruction process have higher SERS EF than AgNPr colloids. This result is consistent with our previous reports [35,36].

## 4. Conclusions

This work studied the photo-induced shape conversion of quasi-spherical nanoparticles to nanoprisms and decahedral NPs in varying bath temperatures under blue LEDs. We found that the temperature affects the shape conversion rates and influences the morphology of the final products. Shape conversion reaction at higher temperatures forms nanoprisms, while decahedral NPs form at lower temperatures. Decahedral silver NP colloids synthesized at 0 °C were not stable, i.e., their edges and tips became blunt within 30 min at ambient temperature. Blunt D-AgNP colloids can be used as seeds for the reconstruction process to rebuild complete D-AgNP colloids under blue LED irradiation and enlargement processes under green LED irradiation. Both small D-AgNP colloids and large D-AgNP colloids exhibited much higher SERS enhancement factors than AgNPr colloids based on the SERS measurements of CV.

## Data Availability

All data generated and analyzed during this study are included in this paper and the attached Appendix A.

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
