# Peer review of "Physical Properties and the Reconstruction of Unstable Decahedral Silver Nanoparticles Synthesized Using Plasmon-Mediated Photochemical Process"

_nanomaterials, 2022, doi:10.3390/nano12071062_

Round 1

Reviewer 1 Report

The authors study the formation and shape transformation of silver nanoparticles (NPs) from solution under blue light varying temperature (T) conditions. They find the formation of prism and decahedral NPs with different distributions depending on T. The stability of decahedral NPs at low temperatures is scrutinized. NP reconstruction at higher temperatures yields not only more stable decahedrals, but also allows to grow the NPs larger under illumination at a different wavelengths.

This experimental study is highly informative and methodically sound. The article is very well written with dense and concise insights into experimental conditions and results, highly suitable for Nanomaterials and its readership. I would recommend its publication with minor changes.

Here some suggestions:
(I) I was wondering about oxidization of silver. Are aging effects observed for the samples?

(II) In the introduction, you describe the reconstruction, but do not mention the ~30 min time frame for this process as stressed in the conclusions. This is an important information to consider together with the other procedures, so I would recommend to add it. Similarly in 2.3.2 when you mention the decay process for the edges of decahedral NPs to become blunt, I would add the time this process takes.

(III) Some graphs are rather small in the text and the labels might be difficult to read. I first thought this about Fig. 4 and would add 5, 7, and 9.

(IV) In the first paragraph of 3.4 the degree signs for temperatures are wrongly formatted.

Author Response

Journal: Nanomaterials
Ms. Ref. No.: nanomaterials-1629600
Title: " Physical Properties and the Reconstruction of Unstable Decahedral Silver Nanoparticles Synthesized Using Plasmon-Mediated Photochemical Process "
Author(s): Jui-Chang Chen, Yu Te Chu, Shi-Hise Chang, Ya-Tin Chuang, and Cheng-Liang Huang*

Dear Reviewer:

     We want to thank you for a review of our manuscript (Manuscript number: nanomaterials-1629600) entitled "Physical Properties and the Reconstruction of Unstable Decahedral Silver Nanoparticles Synthesized Using Plasmon-Mediated Photochemical Process."

We have taken your comments into account and revised the manuscript accordingly. They are all incorporated in the revised manuscript and are marked in red Palatino Linotype. The following is a point-by-point reply to your comments (in green color).

The comments and authors' response:

The authors study the formation and shape transformation of silver nanoparticles (NPs) from solution under blue light varying temperature (T) conditions. They find the formation of prism and decahedral NPs with different distributions depending on T. The stability of decahedral NPs at low temperatures is scrutinized. NP reconstruction at higher temperatures yields not only more stable decahedrals, but also allows to grow the NPs larger under illumination at a different wavelengths.

This experimental study is highly informative and methodically sound. The article is very well written with dense and concise insights into experimental conditions and results, highly suitable for Nanomaterials and its readership. I would recommend its publication with minor changes.

Here some suggestions:
(I) I was wondering about oxidization of silver. Are aging effects observed for the samples?

Response: Thank you very much for the kind question. For the silver nanoprisms synthesized at higher temperatures or decahedral silver nanoparticles rebuilt at higher temperatures, their LSPR spectra can remain almost unchanged for a very long time, i.e., more than two years. This spectroscopic observation indicates that these silver nanoparticles retain their original form after preparation in the water solution. Therefore, the citrate ions in the colloidal solution probably provide the surface charge for the Coloumb repulsive potential and act as reducing agents to prevent the oxidation of AgNPs.  

(II) In the introduction, you describe the reconstruction, but do not mention the ~30 min time frame for this process as stressed in the conclusions. This is an important information to consider together with the other procedures, so I would recommend to add it. Similarly in 2.3.2 when you mention the decay process for the edges of decahedral NPs to become blunt, I would add the time this process takes.

Response: Thank you very much for the thoughtful suggestion. First, we should address the decay time of D-AgNPs synthesized at a low bath temperature. We have rewritten our sentences. In the Introduction section, the sentence was rewritten as "The tips and edges of decahedral silver NPs synthesized by this method at such a low temperature will undergo a decay process to become blunt after removing the light for 30 minutes." In the 2.3.2 section, the sentence was rewritten as "The corners and edges of D-AgNPs would decay and become blunt within 30 minutes."

(III) Some graphs are rather small in the text and the labels might be difficult to read. I first thought this about Fig. 4 and would add 5, 7, and 9.

Response: Thank you for the thoughtful consideration. According to your suggestion, we have enlarged Figures 4, 5, 7, and 9 with suitable sizes.

(IV) In the first paragraph of 3.4 the degree signs for temperatures are wrongly formatted.

Response: Thanks for your correction. According to your suggestion, the wrongly formatted degree signs for temperatures were corrected as the correct format.

In short, we have carefully considered your comments and have revised the manuscript according to their suggestions. We hope you will find our manuscript is now acceptable for publication. Thank you very much for your kind help.

Best regards,

Cheng-Liang Huang, Ph.D.

Professor

Dept. of Applied Chemistry

National Chiayi University

No.300 Syuefu Rd., Chiayi City 60004, Taiwan

Phone: 886-5-2717963

Fax: 886-5-2717901

Reviewer 2 Report

Growth and morphology of nanoparticles develop at different precursor concentrations is crucial. What information is available on the concentration of free (unreacted precursor) during each growth phase? How can be determine these concentration conditions?

According the manuscript: "Powder X-ray diffraction (XRD) was performed on a Bruker D8 Discover
X-ray diffractometer with a Cu K radiation source, and the sample was deposited onto a
glass slide." Where can we find the resuts of the powder X-ray diffraction measurements?

Author Response

Journal: Nanomaterials
Ms. Ref. No.: nanomaterials-1629600
Title: " Physical Properties and the Reconstruction of Unstable Decahedral Silver Nanoparticles Synthesized Using Plasmon-Mediated Photochemical Process "
Author(s): Jui-Chang Chen, Yu Te Chu, Shi-Hise Chang, Ya-Tin Chuang, and Cheng-Liang Huang*

Dear Reviewer:

     We would like to thank you for a review of our manuscript (Manuscript number: nanomaterials-1629600) entitled “Physical Properties and the Reconstruction of Unstable Decahedral Silver Nanoparticles Synthesized Using Plasmon-Mediated Photochemical Process”.

We have taken your comments into account and revised the manuscript accordingly. They are all incorporated in the revised manuscript and are marked in red Palatino Linotype. The following is a point-by-point reply to your comments (in green color).

Your comments and our response:

(I) Growth and morphology of nanoparticles develop at different precursor concentrations is crucial. What information is available on the concentration of free (unreacted precursor) during each growth phase? How can be determine these concentration conditions?

Response: Thank you for this thoughtful consideration. If the ICP-mass is available, the concentration of free silver ions can be determined when AgNPs can be removed by centrifugation. In the daily lab, we have developed an indirect method based on the titration of 16-MHDA (which were not published yet). Figure R1 shows UV-vis spectra (a) and the LSPR redshifts (b) of D-AgNP colloids in the presence of different concentrations of 16-MHDA. The LSPR band of AgNPs redshifts due to forming a self-assembly monolayer (caused the environmental refractive index change). However, in the presence of free silver ions, we need to add more 16-MHDA because 16-MHDA will react with free silver ions before adsorption to AgNPs. Accordingly, the concentration of free silver ions can be determined roughly by the titration of 16-MHDA.

Figure R1. (a) UV-vis spectra and (b) the LSPR redshifts of D-AgNP colloids in the presence of different concentration of 16-MHDA.

(II) According the manuscript: "Powder X-ray diffraction (XRD) was performed on a Bruker D8 Discover X-ray diffractometer with a Cu K radiation source, and the sample was deposited onto a glass slide." Where can we find the resuts of the powder X-ray diffraction measurements?

Response: Thank you very much for the kind suggestion. We have added a figure about XRD spectra of D-AgNPs rebuilt at 60 and 10 ℃ (as shown in Figure S2 in revised manuscript).

In short, we have carefully considered your comments and have revised the manuscript according to their suggestions. We hope you will find our manuscript is now acceptable for publication. Thank you very much for your kind help.

Best regards,

Cheng-Liang Huang, Ph.D.

Professor

Dept. of Applied Chemistry

National Chiayi University

No.300 Syuefu Rd., Chiayi City 60004, Taiwan

Phone: 886-5-2717963

Fax: 886-5-2717901

Reviewer 3 Report

This is a simple but interesting work, correlating the particles growth parameters (temperature, time, exposition to blue or green light) with the morphology of silver nanoparticles. In my opinion, this manuscript should be publishable after some revision.

1) Figure 1: please use the same wavelength axis range for all panels, to facilitate the comparison.

2) Page 6, before Fig. 4: "The exponential curve could correlate to a Boltzmann-like distribution function related to the distribution of two nanostructures with different Gibbs free energy at different temperatures." could the author provide some more accurate information about the mentioned fit (exponential curve) and calculations giving rise to the mentioned "two ....different Gibbs free energy values"? in Fig.4 only experimental values are shown (no fit, no calculations).

3) Page 7, line 5: "unstable property". I would write "the observed instability", since this is not a peculiar chemico-physical property of the nanosystem, but an observed behavior in selected experimental conditions.

4) This is my main point of concern: a great part of the conclusions that the authors report about the process of formation, decay, reconstruction, and enlargement of D-AgNPs (as resumed in Scheme 1) are based on the analysis of TEM images; however, mean particles sizes are reported in the text without a range of confidence. Did the author perform a statistical analysis on the particles dimensions estimated by the TEM images? If not, I strongly suggest to do it. If yes, I suggest to include the statistical analysis results at least in the supporting material.

Author Response

Journal: Nanomaterials
Ms. Ref. No.: nanomaterials-1629600
Title: " Physical Properties and the Reconstruction of Unstable Decahedral Silver Nanoparticles Synthesized Using Plasmon-Mediated Photochemical Process "
Author(s): Jui-Chang Chen, Yu Te Chu, Shi-Hise Chang, Ya-Tin Chuang, and Cheng-Liang Huang*

Dear Reviewer:

     We would like to thank you for a review of our manuscript (Manuscript number: nanomaterials-1629600) entitled “Physical Properties and the Reconstruction of Unstable Decahedral Silver Nanoparticles Synthesized Using Plasmon-Mediated Photochemical Process”.

We have taken your comments into account and revised the manuscript accordingly. They are all incorporated in the revised manuscript and are marked in red Palatino Linotype. The following is a point-by-point reply to your comments (in green color).

Authors’ Replies to Your Comments

This is a simple but interesting work, correlating the particles growth parameters (temperature, time, exposition to blue or green light) with the morphology of silver nanoparticles. In my opinion, this manuscript should be publishable after some revision.

  • Figure 1: please use the same wavelength axis range for all panels, to facilitate the comparison.

Response: Thank you very much for your suggestion. We have corrected Figure 1 (a)-(f) with the same spectral region (as shown below).

Figure 1 Time-dependent UV-vis spectra of silver colloids synthesized at (e) 10 °C, and (f) 0 °C.

  • Page 6, before Fig. 4: "The exponential curve could correlate to a Boltzmann-like distribution function related to the distribution of two nanostructures with different Gibbs free energy at different temperatures." could the author provide some more accurate information about the mentioned fit (exponential curve) and calculations giving rise to the mentioned "two ....different Gibbs free energy values"? in Fig.4 only experimental values are shown (no fit, no calculations).

Response: Thank you very much for your considered opinion. Since we worry that we have over-interpreted our data in Figure 4, we removed these two sentences from the revised manuscript.  

  • Page 7, line 5: "unstable property". I would write "the observed instability", since this is not a peculiar chemico-physical property of the nanosystem, but an observed behavior in selected experimental conditions.

Response: Thank you very much for your thoughtful suggestion. The sentence on Page 7 was rewritten as “This observed instability was not detected for the D-AgNPs synthesized using other methods in the previous studies because these as-prepared D-AgNPs lacked an annealing process when synthesized at low temperatures.”

  • This is my main point of concern: a great part of the conclusions that the authors report about the process of formation, decay, reconstruction, and enlargement of D-AgNPs (as resumed in Scheme 1) are based on the analysis of TEM images; however, mean particles sizes are reported in the text without a range of confidence. Did the author perform a statistical analysis on the particles dimensions estimated by the TEM images? If not, I strongly suggest to do it. If yes, I suggest to include the statistical analysis results at least in the supporting material.

Response: Thank you very much for the considered opinion. As shown in Figure S3 (revised manuscript), we have made a histogram of particle sizes. The number of D-AgNPs in different processes we counted is more than 100. From this more detailed statistical analysis, the average edge lengths of D-AgNPs are 40.8 ± 5.1 nm and 53.3 ± 4.0 nm for the reconstruction under blue LED at 60 ℃ and enlargement process, respectively.

Figure S3. (a) Edge length distribution of D-AgNPs after the reconstruction under blue LEDs at 60 ℃. (b) Edge length distribution of D-AgNPs after the enlargement process.   

In short, we have carefully considered your comments and have revised the manuscript according to their suggestions. We hope you will find our manuscript is now acceptable for publication. Thank you very much for your kind help.

Best regards,

Cheng-Liang Huang, Ph.D.

Professor

Dept. of Applied Chemistry

National Chiayi University

No.300 Syuefu Rd., Chiayi City 60004, Taiwan

Phone: 886-5-2717963

Fax: 886-5-2717901

Round 2

Reviewer 2 Report

Accept in present form

Reviewer 3 Report

The authors carefully considered the suggested revisions; in particular, they considerably improved the presentation and statistical analysis of TEM images, adding histograms in the supporting information (as suggested).

In my opinion, this paper is now publishable on Nanomaterials.